# Determinants of Willingness of Patients with Type 2 Diabetes Mellitus to Receive the Seasonal Influenza Vaccine in Southeast China

**DOI:** 10.3390/ijerph16122203

**Published:** 2019-06-21

**Authors:** Wei Feng, Jun Cui, Hui Li

**Affiliations:** 1Department of Chronic Diseases and Community Health, Fenghua District Center for Disease Control and Prevention, Ningbo 315000, Zhejiang, China; fhcdc01@126.com; 2Department of Chronic Disease Prevention, Ningbo Municipal Center for Disease Control and Prevention, Ningbo 315000, Zhejiang, China; swy_cyq@163.com

**Keywords:** determinants, diabetic patients, seasonal influenza vaccine, willingness

## Abstract

Willingness of patients with Type 2 diabetes mellitus (T2DM) to receive the seasonal influenza vaccine is low in China. A cross-sectional study on a representative sample of T2DM patients was conducted in Ningbo, a city in southeast China, to assess T2DM patients’ willingness to be vaccinated against influenza and identify the influence factors of this willingness. Data regarding the participant’s history of influenza, the knowledge, willingness and uptake of the influenza vaccine, demographic characteristics, reasons for willingness or unwillingness to be vaccinated was collected. Only 19.55% of a total of 1749 participants reported a willingness to be vaccinated. Factors positively associated with willingness to be vaccinated were perceived susceptibility to influenza (OR = 1.9, 95% CI: 1.5–2.5), awareness of the vaccine (OR = 1.7, 95% CI: 1.3–2.3) and previous history of influenza vaccination (OR = 4.4, 95% CI: 3.0–6.4). Patients with T2DM who were farmers (OR = 0.6, 95% CI: 0.4–0.8) and those managed by contracted family doctors (OR = 0.8, 95% CI: 0.6–1.0) expressed less willingness to be vaccinated. Targeted interventions such as enhancing health education and strengthening medical staff training should be conducted to increase T2DM patients’ willingness to be vaccinated and enhance influenza vaccine uptake among this population.

## 1. Introduction

Influenza, a common acute respiratory infection, has become a major public health problem worldwide [1,2]. Evidence from the World Health Organisation (WHO) shows that influenza causes 3–5 million serious illnesses annually [3]. Patients with Type 2 Diabetes Mellitus(T2DM) are generally susceptible to influenza and some also have a metabolic syndrome or cardiovascular disease [4,5], which results in a higher risk of hospitalisation or death due to influenza infection than for healthy individuals [6,7]. In recent years, the prevalence of diabetes has increased rapidly in China from 2.5% in 1994 to 10.9% in 2013, which means that the number of patients with T2DM in the country has reached an estimated 150 million [8,9]. Therefore, patients with T2DM are a key group that we should target for influenza prevention.

Currently, one of the most effective measures to prevent influenza in patients with T2DM is to receive a seasonal influenza vaccine. Several studies have identified influenza vaccination as an essential method to reduce the incidence of influenza complications, including influenza-related hospitalisation and death [10,11,12,13]. Furthermore, both domestic and international guidelines recommend influenza vaccines for people with T2DM [14,15,16,17,18]. The influenza vaccination rate among patients with T2DM reported in China ranged from 2.9% to 28.8% and community-based interventions could significantly increase coverage of influenza vaccine [19,20,21]. Similar result was discovered in Ningbo: the vaccination rate among 1682 patients with T2DM reached 80.0% after adopting community interventions in Dongqianhu Town. However, monitoring results showed a low influenza vaccination rate of only 6.0% among patients with T2DM in non-Dongqianhu Town areas of Ningbo City in 2017. In order to improve the influenza vaccine uptake, primary care physicians in Ningbo City are required to recommend vaccination to patients with T2DM. With the aim of providing a basis for improving future influenza vaccination coverage in this population, our study explored the willingness to receive the influenza vaccine and their influencing factors among patients with T2DM in Ningbo City.

## 2. Materials and Methods

### 2.1. Study Area

Ningbo is a developed city in southeast China with 10 districts and counties and approximately 8.5 million permanent residents. A total number of 240,343 patients in the city attend a free, government-provided diabetes health management program, which was a program offered under the Chinese basic public health service. The program included blood glucose monitoring, health education, lifestyle intervention, health examination, and treatment of diseases for patients aged over 35 with T2DM.

### 2.2. Study Design and Sampling Method

This cross-sectional study targeted patients with T2DM attending the diabetes health management program in Ningbo City from December 2016 to January 2017. The minimum sample size was determined to be 1536 by using a sample size formula. Considering that some patients might refuse to visit the programme, the sample size was expanded by 10%, thus resulting in a final target sample of 1690 individuals.

We used a two-stage sampling method to select research subjects. First, two towns were randomly selected in each of nine counties or districts; one district was excluded because a similar investigation had previously been conducted there. Next, 100 patients with T2DM attending the diabetes health management program in each designated town were randomly selected as research participants. Eventually, a total of 1800 patients with T2DM were invited to participate in our survey, 1784 of them agreed and 1749 subjects completed the questionnaires with response rate of 97.17%. The top two reasons for non-completion of the questionnaire were that 35 subjects refused to answer some questions included in the questionnaire and 16 subjects refused to participate in our investigation.

### 2.3. Ethics Statement

The Ethics Committee of the Ningbo Municipal Center for Disease Control and Prevention approved this research (No.201811). Oral consent was obtained from all participants before their inclusion in the study and participation was anonymous.

### 2.4. Data Collection

Data was collected from selected patients with T2DM through face-to-face interviews by trained investigators. The interview questionnaire consisted of four sections and a total of 34 questions. Section one included 11 questions regarding the subject’s history of influenza and the knowledge of the influenza. Section two contained 6 questions about the participants’ perception of the influenza vaccine. In section three, 7 questions were used to assess the participants’ willingness and uptake of the influenza vaccine. In section four, the participants were asked 10 questions about their demographic characteristics. Most of the questions, such as participants’ knowledge, willingness and practice of influenza vaccine were single-choice. The permissible response options were 1 = ”yes”, 2 = ”no”, 99 = ”unclear”. Few questions like reasons for willingness or unwillingness to receive influenza vaccines were multiple choice. The permissible response options ranged from 1 to 9; the details are displayed in Table 2.

### 2.5. Statistical Analysis

All data were entered using EpiData 3.0 (Odense, Denmark) and analysed applying SPSS Version 13.0 (SPSS Inc., Chicago, IL, USA). Descriptive statistics like frequency and percentages were calculated for categorical variables. A multivariable logistic regression analysis was utilised and odds ratio with a 95% confidence interval (95% CIs) were computed to identify determinants of willingness among patients with T2DM to receive the seasonal influenza vaccine. A two-tailed *p* value < 0.05 was judged as statistically significant.

## 3. Results

### 3.1. Sociodemographic Characteristics of Study Participants

A total of 1800 patients with T2DM were invited for our study; 1784 of them agreed and 1749 participants completed the questionnaires. Among all included subjects, 1351 (77.2%) were aged over 60, 1076 (61.5%) were females, 956 (54.7%) were from urban areas, and 898 (51.3%) were farmers. Approximately three-fourths (75.3%) of the participants earned a monthly income of less than 3000 yuan, and the proportion of patients with T2DM receiving management from a contracted family doctor was 63.0%. Sociodemographic characteristics of the recruited participants are displayed in Table 1.

### 3.2. Reasons for Willingness and Unwillingness to Receive Seasonal Influenza Vaccination among Patients with T2DM

Among the 1749 participants, 342 (19.6%) expressed a willingness to receive the influenza vaccine and 169 (9.7%) participants reported having been previously vaccinated. Reasons for willingness or unwillingness among patients with T2DM to receive the seasonal influenza vaccine are presented in Table 2. The top three reasons reported among 342 willing participants were the vaccine’s effectiveness in reducing the risk of influenza (68.4%), belief in the safety and reliability of vaccines (30.4%), and the free cost or full reimbursement by their medical insurers (27.8%). In contrast, feeling in good health with no need for vaccinations (36.2%), fear of adverse reactions (19.0%), and the high cost of vaccines (17.1%) were the most commonly reported reasons among 1407 unwilling respondents.

### 3.3. Factors Associated with the Willingness to Receive the Seasonal Influenza Vaccine among Patients with T2DM

Table 3 presents the results of the multivariable logistic regression analysis of factors influencing T2DM patients’ willingness to receive the seasonal influenza vaccine. After controlling for other confounding factors, perceived susceptibility to influenza, awareness of the influenza vaccine and previous history of influenza vaccination were positively correlated with the willingness to receive the vaccine. Conversely, the farming occupation and being managed by a contracted family doctor were negatively associated with T2DM patients’ willingness to be vaccinated against influenza.

## 4. Discussion

In the present study, it was found that less than 20% of patients with T2DM were willing to be vaccinated against influenza, which was significantly lower than that observed in the Chinese city of ShenZhen [19]. The proportion of participants who had received the vaccine was even lower at slightly under 10%. Influenza vaccination rates in Ningbo City are significantly lower than those reported in the Netherlands (74.8%) [22], Spain (65.7%) [23], Saudi Arabia (61.2%) [24], and South Korea (50.0%) [25]. In Ningbo City, influenza vaccination rates among patients with T2DM only in Dongqianhu Town exceeded 75%. Consequently, there is a need to carry out targeted interventions to increase the willingness of diabetic patients in Ningbo to be vaccinated and eventually improve the poor vaccination status for influenza among this population.

Our study revealed that the effectiveness of the vaccine to reduce the risk of influenza is the most important factor influencing the willingness of patients with T2DM to be vaccinated. A systematic review published in 2018 indicated that influenza vaccination reduced the risk of hospitalisation and death in patients with T2DM and the cost of hospitalisation was $1283 lower for participants in the vaccination group than for those in the non-vaccination group [11]. There have been several guidelines recommending influenza vaccination for patients with T2DM [14,15,16,17,18], and many countries have incorporated influenza vaccines into their national immunisation plans [26]. An assessment of the cost-effectiveness of influenza vaccination for patients with T2DM in Turkey demonstrated that an increase in the vaccination rate from 9.1% to 20% was cost-effective according to WHO guidelines [27]. In accordance with the recognition of its benefits for patients with T2DM, the city of Ningbo began in 2016 to promote the influenza vaccine for this population.

In our study, self-identification as being in good physical condition and fear of vaccines’ side effects were the two most common reasons why patients with T2DM were reluctant to receive influenza vaccines. This result is consistent with other findings [28,29], and it suggests that some patients with T2DM lack an understanding of the vaccine and have a low awareness of the benefits of influenza vaccination. The supposed high price of vaccines was another commonly-cited reason for T2DM patients’ unwillingness. In 2014, a policy was issued in Ningbo that personal health insurance accounts could be used to reimburse the cost of influenza vaccination for individuals and their families; however, many patients with T2DM remain unaware of this policy. Therefore, medical staff should take advantage of the time spent with patients during visits and follow-ups to carry out targeted health education as a means to promote the vaccine and spread knowledge of the regional reimbursement policy.

Previous studies have demonstrated that recommendations by medical staff are a significant factor in increasing influenza vaccination coverage [30,31,32]. Conversely, in our study, a negative correlation was revealed between management by contracted family doctors and patients’ willingness to be vaccinated, thus indicating a need for family doctors to play a more active role in promoting influenza vaccination among patients with T2DM. The results of a recent survey in Ningbo on whether medical personnel were willing to recommend influenza vaccines to patients with T2DM showed such willingness among only 58.13% of the respondents [33]. However, the proportion of clinicians with a willingness to make such recommendations might be even lower in practice, which might be related to several factors. Firstly, doctors in China are heavily burdened in that they provide outpatient services for a large number of patients daily. Due to time limitations, doctors do not have many opportunities to recommend influenza vaccines to patients with T2DM. Secondly, the concept of disease prevention has not yet been fully established among local clinicians; thus, many medical personnel are primarily concerned with disease treatment and are relatively ignorant of the benefits of recommending influenza vaccines to patients with T2DM. Thirdly, concerns regarding possible side-effects make some medical personnel reluctant to recommend the influenza vaccine to patients with T2DM. Strengthening the training of medical staff, particularly family doctors, is a means to address this situation. In addition, there is a need to gradually increase the number of medical staff to reduce their burden and provide them with sufficient time to recommend influenza vaccine to patients with T2DM.

Compared with other occupations, farmers are more reluctant to be vaccinated against influenza, which might be attributable to their long-term physical work and self-evaluation of being in good physical condition. However, the low health literacy of farmers and insufficient medical resources in rural areas might also result in their lower willingness to receive vaccination. Accordingly, related health education and interventions for farmers should be strengthened.

Other studies have identified a history of prior influenza vaccination as a positive predictor of T2DM patients’ willingness to repeat the experience [34,35,36,37]. Our findings align with those results, as a previous history of influenza vaccination had the highest OR value (4.41) among all factors that were positively related to T2DM patients’ willingness to be vaccinated. In addition, we observed a positive correlation between T2DM patients’ feelings of susceptibility to influenza and their willingness to be vaccinated, which was also consistent with previous findings [38]. Patients with T2DM who are consciously susceptible to influenza not only have greater knowledge of the disease, but also pay more attention to their own physical condition, thus resulting in a relatively higher willingness to be vaccinated.

Our study has several limitations. Firstly, causal relationships are impossible to determine due to the study’s cross-sectional design. Secondly, the subjects of this study were limited to patients with T2DM participating in a diabetes health management program; thus, the findings cannot claim to be representative of all diabetics. Thirdly, recall bias is an inevitable factor, as some indicators reflect information remembered from the past year or earlier, which might influence the percentage of factors associated with T2DM patients’ willingness in our study. Fourth, this article focuses only on T2DM patients’ willingness to receive the influenza vaccination; however, a willingness to be vaccinated does not necessarily lead to the actual practice of vaccination. Further study is warranted to explore the vaccination situation of patients with T2DM who are willing to receive the influenza vaccine.

## 5. Conclusions

This study found a low proportion of patients with T2DM who were willing to be vaccinated against influenza. Self-evaluations of good physical condition, fear of adverse reactions, and the perceived high cost of vaccines were the main barriers to vaccination, whereas perceived susceptibility to influenza, awareness of the vaccine, and a previous history of influenza vaccination increased T2DM patients’ willingness to be vaccinated. T2DM patients who were farmers and those being managed by family doctors expressed less willingness to be vaccinated. Targeted interventions such as enhancing health education for diabetic patients and strengthening the training of medical personnel should be conducted to effectively promote influenza vaccination among diabetic patients.

## Figures and Tables

**Table 1 ijerph-16-02203-t001:** Sociodemographic characteristics of study participants (N = 1749).

Variable	Participants	Proportion (%)
Age(years)		
<60	398	22.8
≥60	1351	77.2
Gender		
Male	673	38.5
Female	1076	61.5
Region		
Urban	956	54.7
Rural	793	45.3
Occupation		
Farmer	898	51.3
Others	851	48.7
Educational Level		
Primary School or Below	1167	66.7
Junior-senior High School or Above	582	33.3
Marital status		
Married	1509	86.3
Divorce or others	240	13.7
Monthly income		
<3000 ¥	1317	75.3
≥3000 ¥	432	24.7
Management by contracted family doctor		
No	647	37.0
Yes	1102	63.0

**Table 2 ijerph-16-02203-t002:** Reasons for willingness and unwillingness among patients with T2DM to receive seasonal influenza vaccination (multiple choice).

Reasons for willingness to be vaccinated against influenza	N	%	Reasons for unwillingness to be vaccinated against influenza	N	%
Effectiveness of the vaccine to reduce the risk of influenza	234	68.4	In good health with no need for vaccination	509	36.2
Safe and reliable vaccines	104	30.4	Fear of adverse reactions	267	19.0
Free charge or full reimbursement by medical insurance	95	27.8	High cost of vaccines	241	17.1
Feelings of susceptibility to influenza	90	26.3	Poor effect of influenza vaccine	104	7.4
Doctor’s advice	83	24.2	No advice from doctors	96	6.8
Friends’ or family members’ advice	27	7.9	Inconvenient vaccination	92	6.5
Suffered from influenza last year	18	5.3	No knowledge of the time or place of vaccination	68	4.8
Other	4	1.2	Other	49	3.5
			Vaccination contraindication	22	1.6

**Table 3 ijerph-16-02203-t003:** Multivariable logistic regression analysis for influential factors associated with T2DM patients’ willingness to receive seasonal influenza vaccination.

Characteristic		Willingness to be Vaccinated	Unwillingness to be Vaccinated	OR (95%CI)	*P* Value
		No (%)	No (%)		
Gender	Female	194 (18.0)	882 (82.0)	1	
	Male	148 (22.0)	525 (78.0)	1.1 (0.9–1.5)	0.34
Region	Rural	130 (16.4)	663 (83.610)	1	
	Urban	212 (22.2)	744 (77.8)	1.2 (0.9–1.6)	0.12
Occupation	Others	207 (24.3)	644 (75.7)	1	
	Farmer	135 (15.0)	763 (85.0)	0.6 (0.4–0.8)	<0.01
Educational level	Primary school or below	211 (18.1)	956 (81.9)	1	
	Junior-senior high school or above	131 (22.5)	451 (77.5)	1.1 (0.8–1.6)	0.42
Age	<60	64 (16.1)	334 (83.9)	1	
	≥60	278 (20.6)	1073 (79.4)	1.4 (1.0–1.9)	0.06
Management by contracted family doctor	No	131 (20.2)	516 (79.8)	1	
	Yes	211 (19.2)	891 (80.8)	0.8 (0.6–1.0)	0.04
Marital status	Divorced or other	44 (18.3)	196 (81.7)	1	
	Married	298 (19.8)	1211 (80.2)	1.1 (0.8–1.7)	0.49
Monthly income	<3000 ¥	245 (18.6)	1072 (81.4)	1	
	≥3000 ¥	97 (22.5)	335 (77.5)	1.3 (0.9–1.8)	0.17
Previous history of influenza vaccination	No	251 (15.9)	1329 (84.1)	1	
	Yes	91 (53.8)	78 (46.2)	4.4 (3.0–6.4)	<0.01
Perceived susceptibility to influenza	No	218 (16.3)	1120 (83.7)	1	
	Yes	124 (30.2)	287 (69.8)	1.9 (1.5–2.5)	<0.01
Awareness of influenza vaccine	No	138 (13.6)	876 (86.4)	1	
	Yes	204 (27.8)	531 (72.2)	1.7 (1.3–2.3)	<0.01
Awareness of vaccination reimbursement policy	No	288 (18.4)	1273 (81.6)	1	
	Yes	54 (28.7)	134 (71.3)	1.2 (0.8–1.7)	0.48

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
