# Peer review of "Determinants of Willingness of Patients with Type 2 Diabetes Mellitus to Receive the Seasonal Influenza Vaccine in Southeast China"

_ijerph, 2019, doi:10.3390/ijerph16122203_

Round 1

Reviewer 1 Report

In this cross-sectional study, the authors have interviewed patients with diabetes, who were attendees of a diabetes health management program in Ningbo, China, to identify factors associated with willingness to receive seasonal infuenza vaccine. The report that prior vaccination, perceived susceptibility to flu-related illness, and awareness of influenza vaccine were positively associated with willingness to be vaccinated. Being farmers and visiting a contracted family doctor for care were negatively associated. They recommend health education for patients with diabetes and more training for medical staff to improve willingness and vaccine uptake rates in this community.

General Comments:

"Diabetic Patient" vs "Patients with diabetes".  In recent times there is growing trend  recommending  to place the person first before disease.  (Ref: Dickinson JK. Commentary: The Effect of Words on Health and Diabetes. Diabetes Spectr. 2017;30(1):11–16. doi:10.2337/ds15-0054) 

Diabetes health management program, contracted family doctor. An non-Chinese reader may not be familiar with these terms and care must be taken to explain what these terms mean. For example, what is diabetes health management program? What does contracted family doctor mean? Does it mean the study participant is obligated/contracted to seek all care one family care doctor or did authors mean contacted/visited family doctor?

Why was actual vaccination and factors associated with prior year's seasonal vaccination not studied instead of willingness. It is well known that willingness does not necessarily mean vaccine uptake.

The quality of language used in the article needs improvement. I will not highlight each and every grammatical issue as there are many and would highly recommend revision of the write-up with help of professional writers.

Specific comments:

Introduction:

This section presents the burden of disease and context for the study. It doesn't however talk about similar studies on the same topic from China. A quick literature search points to a few papers on similar topic from Chinese provinces. Would recommend addition of summary of current literature and how and why this study is different from other similar studies? At the very least you should have spoken about similar study conducted in 10th county.

Materials and Methods:

Line 55-58. You mention Ningbo to be a developed city. How come a good proportion of your study participants are rural, farmers, and have less than high school education? which is not typical for residents of a developed city. So, by design, have you began with a pool of subjects that are not representative of Ningbo population but rather those who are under-resourced and hence use the free diabetes health management program? This selection bias will pose a threat to generalizability of study findings.

To help readers better understand some of these issues, please address comment 2 under general comments.

Line 69-71. Please ensure language is consistent throughout. Were study patients identified from diabetes health management program sites or from offices of community doctors? Or were they participants of diabetes health management program run from offices of community doctors? Please clarify.

Line 71-72. 2 towns * 9 counties* 100 patients =1800 patients. Did all of them consent to participate and then later only 1749 completed the interviews? Please clarify how many were approached, over what period of time, and what % consented and how many completed interviews. Can you add a line or two with top two-three reasons for non-completion of questionnaire.

Line 78-81. Was data collection anonymous? Were names or interviews recorded? Describe the questionnaire in more detail. How many total questions and in each sub-section and what were the permissible responses (yes/no, likert scales,etc ) ?

Line 84-85. First speak of descriptive statistics.

Line 85-85. Re-frame line regarding purpose of chi-square test.

Line 85. Multivariable logistic regression analysis was performed not multivariate. Multivariate is applicable when outcomes are assessed more than once for the same subject.

Results:

Line 98. In the table title describe n (n=XXX). Describe units of Monthly income in table 1. 

Were only 3 variables tested for intent? It is not clear who is part of Table 2 and Table 3. It appears that who intended to receive were asked questions regarding willingness and those who showed negative intent were asked unwillingness questions? Please clarify as this was not clear reading the methods or results text? What is use of this step if you planned to conduct a multivariable logistic regression with all explanatory variables?

Table 4 title reads as intent but headings in the table read as willingness/unwillingness. It is confusing. Please clarity these terms throughout the paper.

Use of 3 decimal points is not necessary. Just use two decimal points.

Discussion:

Please compare these results with other studies from China and the 10th county in Ningbo.

Comment on how the study results might be affected by various limitations you sight.

Author Response

Reviewer #1:

1)Introduction: This section presents the burden of disease and context for the study. It doesn't however talk about similar studies on the same topic from China. A quick literature search points to a few papers on similar topic from Chinese provinces. Would recommend addition of summary of current literature and how and why this study is different from other similar studies? At the very least you should have spoken about similar study conducted in 10th county.

Responses: Thank you. We added summary of current literature and similar study conducted in Ningbo in line 44-48 in page 2. The difference between this paper and other similar studies are that some influence factors associated with T2DM patients’ willingness is inconsistent, such as vaccination reimbursement policy and management by contracted family doctor.

2) Materials and Methods: Line 55-58. You mention Ningbo to be a developed city. How come a good proportion of your study participants are rural, farmers, and have less than high school education? Which is not typical for residents of a developed city. So, by design, have you began with a pool of subjects that are not representative of Ningbo population but rather those who are under-resourced and hence use the free diabetes health management program? This selection bias will pose a threat to generalizability of study findings.

Responses: Thank you. In Ningbo, there are 5 districts belonging to urban areas and 5 counties belonging to rural areas. According to the research plan, we randomly selected 200 patients from 4 districts and 5 counties who participated in diabetes health management program. Therefore, our results show that 43.54% of the participants are from rural areas, and 51.34% of the participants are rural. In addition, these participants accounted for 77.24% of those over the age of 60, so most of them have less than high school education. The health management services of diabetes patients in urban areas and rural areas are consistent, because this service belongs to The National Basic Public Health Service Project which is provided for free by the government. More than 80% patients attending diabetes health management program aged over 60.

3) To help readers better understand some of these issues, please address comment 2 under general comments.

Responses: Thank you for your suggestions. We address comment 2 under general comments.

Comment 2

3.1 "Diabetic Patient" vs "Patients with diabetes".  In recent times there is growing trend  recommending  to place the person first before disease.  (Ref: Dickinson JK. Commentary: The Effect of Words on Health and Diabetes. Diabetes Spectr. 2017;30(1):11–16. doi:10.2337/ds15-0054) 

Responses: Thank you for your vital comments. We used patients with diabetes as the main expression in our article.

 3.2 Diabetes health management program, contracted family doctor. An non-Chinese reader may not be familiar with these terms and care must be taken to explain what these terms mean. For example, what is diabetes health management program? What does contracted family doctor mean? Does it mean the study participant is obligated/contracted to seek all care one family care doctor or did authors mean contacted/visited family doctor?

Responses: Thank you. Diabetes health management program is one of the program included in project of Chinese basic public health service, which was provided by the government for free. The client is a type 2 diabetes patients aged over 35. The program includes blood glucose monitoring, health education, lifestyle intervention, health examination and treatment of diabetes. Family doctor could provide the service of diabetes health management program so contracted family doctor mean management or served by contracted family doctor. In the paper, we changed contracted family doctor to management by contracted family doctor.

  3.3 Why was actual vaccination and factors associated with prior year's seasonal vaccination not studied instead of willingness. It is well known that willingness does not necessarily mean vaccine uptake.

Responses: Thank you. Prior year's seasonal influenza vaccination rate was less than 1%. If we study influenza vaccination rate, we need a large sample of approximately 39600 individuals. So, firstly we want to increase T2DM patients’ willingness to be vaccinated and then enhance influenza vaccine uptake among this population.

  3.4 The quality of language used in the article needs improvement. I will not highlight each and every grammatical issue as there are many and would highly recommend revision of the write-up with help of professional writers.

Responses: Thank you. We had asked for a highly qualified native English speaker in the field of medicine to revise our manuscript for proper English language, grammar, punctuation and spelling.

4) Line 69-71. Please ensure language is consistent throughout. Were study patients identified from diabetes health management program sites or from offices of community doctors? Or were they participants of diabetes health management program run from offices of community doctors? Please clarify.

Responses: Thank you for your important suggestions. Study patients were identified from diabetes health management program sites. We clarify it in line 74 -76 in page 2.

5) Line 71-72. 2 towns * 9 counties* 100 patients =1800 patients. Did all of them consent to participate and then later only 1749 completed the interviews? Please clarify how many were approached, over what period of time, and what % consented and how many completed interviews. Can you add a line or two with top two-three reasons for non-completion of questionnaire.

Responses: Thank you. In our study, a total of 1800 diabetics were invited to participate in our survey, 1784 of them agreed and 1749 completed the questionnaires with the response rate of 97.17%. This study was conducted in Ningbo City during December 2016 to January 2017. The top two reasons for non-completion of the questionnaire were that 35 subjects refused to answer some questions included in the questionnaire and 16 subjects refused to participate in our investigation. In our paper, we added them in line 76-80, page 2.

6) Line 78-81. Was data collection anonymous? Were names or interviews recorded? Describe the questionnaire in more detail. How many total questions and in each sub-section and what were the permissible responses (yes/no, likert scales,etc ) ?

Responses: Thank you for your suggestions. Yes, data collection was anonymous and participants' names were not recorded. The questionnaire consisted of four sections and a total of 34 questions and we provided more details of survey questionnaire in line 87-96 in page 3.

7) Line 84-85. First speak of descriptive statistics.

Responses: Thank you for this vital advice. We added descriptive statistics in line 99-100 in page 3.

8) Line 85-85. Re-frame line regarding purpose of chi-square test.

Responses: Thank you. We deleted Table 2 and chi-square test was not finally performed in our article.

9) Line 85. Multivariable logistic regression analysis was performed not multivariate. Multivariate is applicable when outcomes are assessed more than once for the same subject.

Responses: Thank you for this vital advice let me understand the difference between multivariable and multivariate. We changed all multivariate to multivariable in our article.

10) Results: Line 98. In the table title describe n (n=XXX). Describe units of Monthly income in table 1.

Responses: Thank you. We describe (N=1749) in the table title in line 114 and add units ¥ of Monthly income in table 1.

11) Were only 3 variables tested for intent? It is not clear who is part of Table 2 and Table 3. It appears that who intended to receive were asked questions regarding willingness and those who showed negative intent were asked unwillingness questions? Please clarify as this was not clear reading the methods or results text? What is use of this step if you planned to conduct a multivariable logistic regression with all explanatory variables?

Responses: Thank you for your vital suggestions. In our article, there were 12 variables tested for willingness. In order to express our results more clearly we delete table 2. Besides, 342 willing participants were asked reasons for willingness to be vaccinated and 1407 unwilling respondents were asked unwillingness questions. The use of this step was to understand motivators and barriers for influenza vaccination among patients with diabetes. After delete table 2, this paragraph had no relationship with the outcome of a multivariable logistic regression.

12)Table 4 title reads as intent but headings in the table read as willingness/unwillingness. It is confusing. Please clarity these terms throughout the paper.

Responses: Thank you. We replace all intent with willingness in our paper.

13)Use of 3 decimal points is not necessary. Just use two decimal points.

Responses: Thank you for your suggestions. We use two decimal points in our article.

14)Discussion:

Please compare these results with other studies from China and the 10th county in Ningbo.

Responses: Thank you. We compare our results with other studies from China and the 10th county in Ningbo in line 144-145 and in line 148-149 in page 6. 

15) Comment on how the study results might be affected by various limitations you sight.

Responses: Thank you. We comment on two limitations. One was the subjects of this study were limited to T2DM patients, thus the findings cannot claim to be representative of all diabetics. The other was recall bias is an inevitable factor, which might influence the percentage of factors associated with T2DM patients’ willingness in our study.

Reviewer 2 Report

This study addresses an important issue, i.e. low coverage with influenza vaccine among patients with type 2 diabetes in Ningbo, China. The study is designed to identify factors that influence willingness to receive vaccine among this population. 

The data collection methods should be made more clear. It appears in Table 3 that patients who indicated willingness to be vaccinated were asked a separate set of questions compared to those who indicated unwillingness to receive vaccine. But Table 4 shows a different set of factors, and these must have been asked of both groups in order to appear as explanatory variables in a multivariable model. 

Abstract: The abbreviation T2DM appears only once in the paper and should be spelled out as Type 2 Diabetes mellitus.

It's not clear what is meant by "contracted family doctor" - perhaps this could be reworded.

In the tables, when the variable "Monthly income" is included, it would be helpful to indicate the unit of currency is the Yuan.

Subject to requirements of the journal, I would recommend rounding all numbers to one digit after the decimal.

Author Response

Reviewer #2:

1)The data collection methods should be made more clear. It appears in Table 3 that patients who indicated willingness to be vaccinated were asked a separate set of questions compared to those who indicated unwillingness to receive vaccine. But Table 4 shows a different set of factors, and these must have been asked of both groups in order to appear as explanatory variables in a multivariable model.

Responses: Thank you. Based on your advice, we had provided more details of the survey questionnaire in line 86-96 in page 3. Table 3(After revised our manuscript, we change Table 3 to Table 2) depicted T2DM patients’ willingness and unwillingness to undergo seasonal influenza vaccination and permissible responses options for this question were multiple choice. 342 participants intended to receive vaccine were asked questions regarding willingness and 1407 participants showed negative intent were asked unwillingness questions. The purpose of this step was to depict motivators and barriers for seasonal influenza vaccination in the study T2DM patients. Table 4 shows a different set of factors and permissible responses options for these factors were single-choice.

2) Abstract: The abbreviation T2DM appears only once in the paper and should be spelled out as Type 2 Diabetes mellitus.

Response: Thank you. We change all diabetes to T2DM because participants recruited in this study were patients with T2DM.

3) It's not clear what is meant by "contracted family doctor" - perhaps this could be reworded.

Responses: Thank you for your vital suggestions. Diabetes health management program is one of the program included in project of Chinese basic public health service, which was provided by the government for free. Family doctor could provide the service of diabetes health management programso contracted family doctor mean management or served by contracted family doctor. In the paper, we changed contracted family doctor to management by contracted family doctor.

4) In the tables, when the variable "Monthly income" is included, it would be helpful to indicate the unit of currency is the Yuan.

Responses: Thank you for your suggestions. We added units ¥ of Monthly income in table 1 and table 3.

5) Subject to requirements of the journal, I would recommend rounding all numbers to one digit after the decimal.

Responses: Thank you for your advice. We use one decimal points in our article except OR(95%CI) and P value in table 3.

Round 2

Reviewer 1 Report

Author comments:

The authors have done a reasonable job in improving the manuscript.

General Comments:

1.      Why was actual vaccination and factors associated with prior year's seasonal vaccination not studied instead of willingness? It is well known that willingness does not necessarily mean vaccine uptake.

The authors notified that in the year prior to the survey that only about 1% had actually received influenza vaccine. My original concern regarding limited correlation between willingness and actual vaccination remains (1 vs 20%) and hence the utility of such analyses as the factors we find significant for “willingness to get vaccinated” may be entirely different from ones associated with “vaccination”.

Specific comments:

Abstract:

Line 12-15: Suggested reframing the sentence as: “A cross-sectional survey of patients with T2DM attending a diabetes health management program was conducted in Ningbo, a city in southeast China. The purpose of the study was to identify factors associated with willingness to be vaccinated against influenza.

Line 15: The word “subject’s” means “of one subject” and is incorrect; can be dropped here.

Line 16: The meaning of the words “practice of influenza vaccine” is not clear. Please clarify or drop them.

Line 19: Suggest replacing “correlation” with “association”. Also, the first part of this sentence needs to be reframed.

Line 20-22: Some of stats here had 3 decimal points; elsewhere 2 and 1 decimal points were used. Please use single decimal throughout the paper (text, tables) for consistency

Introduction:

Line 49-50: Please clarify that 6.7% stat pertains to non-Dongqianhu Town areas of Ningbo City and hence the need for this study.

Line: Please replace “T2DM patients” with “patients with T2DM” throughout the paper

Materials and Methods:

Line 77: Drop word “were”

Line 86: Replace “with” with “by”

Line 87: Suggest saying “interview questionnaire”

Line: Again please clarify “practice of vaccine”

Line 98-104: It is not clear why a few factors listed in Table 2 were not part of the ones used in the regression models in Table 3. Was the factor selection driven by prior literature or was it done ad hoc.

Results:

Line 107-108: Please reframe to indicate 1800 were approached, 1784 consented, and 1749 completed the survey interviews

Table 2 and 3 title: Suggest replacing word “undergo” with “receive”

Discussion:

Line 144: Suggest replacing “conduct” with “observed”

Line 149: Suggest replacing “overpass” with “exceeded”

Author Response

Reviewer #1:

1) General Comments:

 Why was actual vaccination and factors associated with prior year's seasonal vaccination not studied instead of willingness? It is well known that willingness does not necessarily mean vaccine uptake.

The authors notified that in the year prior to the survey that only about 1% had actually received influenza vaccine. My original concern regarding limited correlation between willingness and actual vaccination remains (1 vs 20%) and hence the utility of such analyses as the factors we find significant for “willingness to get vaccinated” may be entirely different from ones associated with “vaccination”.

Responses: Thank you for your suggestions. We also agree with you that the factors we find significant for “willingness to get vaccinated” may be entirely different from ones associated with “vaccination”. In order to improve the influenza vaccine uptake among patients with T2DM, we have two stages about this work. First, we want to increase T2DM patients’ willingness to be vaccinated. Then enhance influenza vaccine uptake among this population through community intervention. This study could provide a basis for our first stage work in the future.

Specific comments:

2) Abstract: Line 12-15: Suggested reframing the sentence as: “A cross-sectional survey of patients with T2DM attending a diabetes health management program was conducted in Ningbo, a city in southeast China. The purpose of the study was to identify factors associated with willingness to be vaccinated against influenza.

Responses: Thank you. We reframed the sentence in line 12 -15 in page 1.

3) Line 15: The word “subject’s” means “of one subject” and is incorrect; can be dropped here.

Responses: Thank you. We changed subject’s to participant's.

4) Line 16: The meaning of the words “practice of influenza vaccine” is not clear. Please clarify or drop them.

Responses: Thank you for your suggestions. We changed “practice of influenza vaccine” to “uptake of influenza vaccine” .

5) Line 19: Suggest replacing “correlation” with “association”. Also, the first part of this sentence needs to be reframed.

Responses: Thank you for your important suggestions. We reframed the first part of this sentence and replaced correlation” with “associated”

6) Line 20-22: Some of stats here had 3 decimal points; elsewhere 2 and 1 decimal points were used. Please use single decimal throughout the paper (text, tables) for consistency

Responses: Thank you. We use single decimal throughout the paper.

7) Introduction: Line 49-50: Please clarify that 6.7% stat pertains to non-Dongqianhu Town areas of Ningbo City and hence the need for this study.

Responses: Thank you for your suggestions. We clarify that influenza vaccination rate 6.1% among patients with T2DM was in non-Dongqianhu Town areas of Ningbo City.

8) Line: Please replace “T2DM patients” with “patients with T2DM” throughout the paper.

Responses: Thank you. We replace “T2DM patients” with “patients with T2DM” throughout the paper.

9) Materials and Methods: Line 77: Drop word “were”

Responses: Thank you. We drop word “were” in line 77.

10) Line 86: Replace “with” with “by”

Responses: Thank you. We replace “with” with “by” in line 86.

11) Line 87: Suggest saying “interview questionnaire”

Responses: Thank you. We change questionnaire” to “interview questionnaire” in line 87.

12) Line: Again please clarify “practice of vaccine”

Responses: Thank you. We change “practice of vaccine” to “uptake of vaccine”

13) Line 98-104: It is not clear why a few factors listed in Table 2 were not part of the ones used in the regression models in Table 3. Was the factor selection driven by prior literature or was it done ad hoc.

Responses: Thank you for your suggestions. Table 2 depicted T2DM patients’ willingness and unwillingness to undergo seasonal influenza vaccination and permissible responses options for this question were multiple choice. 342 participants intended to receive vaccine were asked questions regarding willingness and 1407 participants showed negative intent were asked unwillingness questions. The purpose of this step was to depict motivators and barriers for seasonal influenza vaccination in the study T2DM patients. Table 3 shows a different set of factors and permissible responses options for these factors were single-choice. Some factors were selected by prior literature and some factors are based on the actual situation of Ningbo City.

14) Results: Line 107-108: Please reframe to indicate 1800 were approached, 1784 consented, and 1749 completed the survey interviews.

Responses: Thank you. We reframe the sentence to indicate 1800 were approached, 1784 consented, and 1749 completed the survey interviews.

15) Table 2 and 3 title: Suggest replacing word “undergo” with “receive”

Responses: Thank you. We replace word “undergo” with “receive” in Table 2 and 3 title.

16) Discussion: Line 144: Suggest replacing “conduct” with “observed”

Responses: Thank you. We replace word conduct” with “observed” in line 144.

17) Line 149: Suggest replacing “overpass” with “exceeded”

Responses: Thank you. We replace word overpass” with “exceeded” in line 149.

Reviewer 2 Report

The methods are now much more clear to me - thank you for the additional information. 

Recommendations / Questions:

-    Page 1, line 16: Could you elaborate on what is meant by "practice of the influenza vaccine"?

-    Page 2, lines 65-71: Remove section on sample size calculations - this degree of detail is              unnecessary.

-    Page 3, line 94: Was 99='unclear' really a response option (i.e. something that the study                subject could have chosen as an answer to a question?)

-    Page 3, line 113: Change "recruited" to "study". 1800 subjects were recruited, but this table          describes characteristics of the 1749 who participated in the study.

-    For all numbers except for p-values, include just 1 digit after the decimal. 

-    Table 1: For monthly income, include a category for missing data, or indicate in a footnote that       54 subjects did not provide this information.

-    I'm still not sure what the relevance of "management by contracted family doctor" is. Could            you explain what the alternative might be and why you chose to include this variable in the            study?

-    Table 3: Recommend switching the "unwillingness to be vaccinated" and the "willingness to be       vaccinated" columns, since it appears that you are modeling willingness to be vaccinated as         the outcome of interest in the multivariable model.

Author Response

Reviewer #2:

1) Page 1, line 16: Could you elaborate on what is meant by "practice of the influenza vaccine"?

Responses: Thank you. W change "practice of the influenza vaccine" to "uptake of the influenza vaccine" in line 16 in page 1.

2) Page 2, lines 65-71: Remove section on sample size calculations - this degree of detail is              unnecessary.

Response: Thank you. We remove section on sample size calculations.

3) Page 3, line 94: Was 99='unclear' really a response option (i.e. something that the study                subject could have chosen as an answer to a question?)

Responses: Thank you for your vital suggestions. More than 80% patients with T2DM attending diabetes health management program aged over 60. So most of them have low educational level. In response to this situation, 99='unclear' was a response option in the questionnaire.

4) Page 3, line 113: Change "recruited" to "study". 1800 subjects were recruited, but this table describes characteristics of the 1749 who participated in the study.

Responses: Thank you for your suggestions. We used the following sentence. A total of 1800 patients with T2DM were invited for our study, 1784 of them agreed and 1749 participants completed the questionnaires.

5) For all numbers except for p-values, include just 1 digit after the decimal.

Responses: Thank you for your advice. We include just 1 digit after the decimal in our paper.

6) Table 1: For monthly income, include a category for missing data, or indicate in a footnote that 54 subjects did not provide this information.

Responses: Thank you for your advice. I am very sorry the figure 1263 is wrongly written. The correct number is 1317 and we revise it in Table 1.

7) I'm still not sure what the relevance of "management by contracted family doctor" is. Could you explain what the alternative might be and why you chose to include this variable in the study?

Responses: Thank you. Diabetes health management program is one of the program included in project of Chinese basic public health service, which was provided by the government for free. The program includes blood glucose monitoring, health education, lifestyle intervention, health examination and treatment of diabetes. Family doctor could provide the service of diabetes health management programso management by contracted family doctor mean treated or served by contracted family doctor. Previous studies have demonstrated that recommendations by medical staff are a significant factor in increasing influenza vaccination coverage. We chose this variable to determine whether medical staff play a role in increasing the willingness of patients with T2DM to be vaccined against influenza.

8)Table 3: Recommend switching the "unwillingness to be vaccinated" and the "willingness to be vaccinated" columns, since it appears that you are modeling willingness to be vaccinated as the outcome of interest in the multivariable model.

Responses: Thank you. We switched the "unwillingness to be vaccinated" and the "willingness to be vaccinated" columns in Table 3.
